# Expression Patterns of Three Important Hormone Genes and Respiratory Metabolism in *Antheraea pernyi* during Pupal Diapause under a Long Photoperiod

**DOI:** 10.3390/insects12080699

**Published:** 2021-08-04

**Authors:** Qi Wang, Yu-Tong Luo, Yong Wang, De-Yi Wang, Xiao-Xia Duan, Yao-Ting Zhang, Yu-Meng Bian, Wei Liu, Li Qin

**Affiliations:** 1College of Bioscience and Biotechnology, Shenyang Agricultural University, Insect Resource Research Center for Engineering and Technology of Liaoning Province, Shenyang 110866, China; wangqi70123@163.com (Q.W.); luoyutong2020@163.com (Y.-T.L.); deyiwang0316@163.com (D.-Y.W.); duanxiaoxia202@163.com (X.-X.D.); bianyumeng614@163.com (Y.-M.B.); liuwei901223@163.com (W.L.); 2Henan Academy of Sericulture Science, Zhengzhou 450008, China; ckyyaoting@126.com

**Keywords:** *Antheraea pernyi*, diapause, prothoracicotropic hormone, ecdysis triggering hormone, eclosion hormone

## Abstract

**Simple Summary:**

In insects, the precise timing of metamorphosis and diapause is regulated by hormones. The Chinese oak silkworm, *Antheraea pernyi*, is a typical pupal diapause insect. Bivoltine species enter diapause in winter and terminate it under suitable environmental conditions in the following year; they produce 70% of total cocoons, whereas univoltine species in lower-latitude areas enter diapause in summer and contribute just one generation a year. A long photoperiod can trigger termination of pupal diapause. It is not clear how photoperiod influences hormone gene expression. Here, hormone-related genes were cloned, and their expression patterns were studied under different photoperiod treatments. The results will help us to understand the molecular changes during diapause termination under long photoperiods and improve breeding of multi-generation tussah pupae in areas where they are naturally univoltine.

**Abstract:**

The Chinese oak silkworm is commonly used in pupal diapause research. In this study, a long photoperiod was used to trigger pupal diapause termination. Genes encoding three hormones, namely prothoracicotropic hormone (PTTH), ecdysis triggering hormone (ETH), and eclosion hormone (EH), were studied. Additionally, ecdysteroids (mainly 20-hydroxyecdysone, 20E) were quantified by HPLC. Pupal diapause stage was determined by measuring respiratory intensity. The pupae enter a low metabolic rate, which starts approximately 1 month after pupal emergence. *ApPTTH* expression showed a small increase at 14 days and then a larger increase from 35 days under the long photoperiod treatment. A similar pattern was observed for the titer of 20E in the hemolymph. However, *ApETH* expression later increased under the long photoperiod treatment (42 days) just before eclosion. Moreover, *ApEH* expression increased from 21 to 35 days, and then decreased before ecdysis. These results suggest that hormone-related gene expression is closely related to pupal development. Our study lays a foundation for future diapause studies in *A. pernyi*.

## 1. Introduction

Diapause—programmed developmental arrest—helps several insects survive in unfavorable environments. Insects can enter diapause in various developmental stages, including the egg, larval, pupal, and adult stages. Diapause involves reduced metabolic activity during adverse seasons, and it can occur in response to seasonal changes in both temperature and day length. Unlike dormancy, diapause does not immediately end under favorable conditions [1]. Insects have to adjust their behavior, metabolism, physiology, and developmental course to adapt to day length [2]. It can regulate energy use and maximize allocation towards growth and reproduction during suitable seasons [3]. Ecdysteroids secreted by the prothoracic glands (PGs) or analogous organs are important for growth, molting, and metamorphosis [4]. Classic ligature and transplantation experiments in lepidopterans have demonstrated that the brain regulates ecdysone production [5]. Prothoracicotropic hormone (PTTH), a key ecodysteroidogenic inducer, has been detected in a pair of lateral neurosecretory cells in the protocerebrum in *Bombyx mori* heads [6]. PTTH stimulates the PGs to synthesize and release ecdysone. In *Drosophila*, PTTH plays a central role in controlling the duration of the larval phase. Knockout of *PTTH* causes several developmental defects [7]. In the cabbage army moth, *Mamestra brassicae*, PTTH is maintained at high levels in non-diapausing pupae, but it is often undetectable in diapausing pupae [8].

Some insect species enter diapause in a fixed stage; diapause is regulated through endocrine hormone release [9]. During diapause, hormonal and neural components regulate insect ecdysis behavior. The brain can release hormones as an initial signal to induce other organ reactions. Peak PTTH titer for a long duration in the hemolymph suggests that PTTH leads to pupa–adult development through PG stimulation in *B. mori* [10]. Ecdysis triggering hormone (ETH) from Inka cells located in the epitracheal glands and eclosion hormone (EH) released by ventromedial (VM) neurosecretory cells in the brain are two master hormones in the pupal ecdysis process [11]. They work together and influence each other. ETH, which is released into hemolymph, can promote VM neurons producing EH. In contrast, EH is involved in the process of ETH release in *Drosophila* and *Manduca* by acting directly on Inka cells to induce ETH release [12,13]. In a previous study, *EH* null mutants of *Drosophila* invariably died during ecdysis and could not be rescued by ETH injection, indicating that EH also plays an essential role in ecdysis [14].

The Chinese oak silkworm (also known as tussah), *Antheraea pernyi* (Guérin-Méneville), a lepidopteran, is widely reared in China. It is an excellent model system for photoperiodic diapause studies [15]. They live in the wild, and their habitat ranges from Heilongjiang and Liaoning to Shandong and Guangxi Provinces [16]. They can adapt to different environments and day lengths via the pupal diapause strategy. However, diapause in the Chinese oak silkworm does not often coordinate with plant growth. In South China, *A. pernyi* pupae are always univoltine under short day lengths. Therefore, many oak trees cannot be used for tussah rearing. Conversely, in North China, most *A. pernyi* pupae are bivoltine. However, in Heilongjiang and Inner Mongolia, trees cannot support larval growth twice a year. Therefore, they are often artificially controlled to ensure that they only breed once a year due to cold weather [17]. Different day lengths influence pupal diapause initiation and termination [16]. Previously, exposure of diapausing pupae of *A. pernyi* to a long photoperiod (16–20 h) terminated diapause and initiated eclosion in almost 80% of the pupae under artificial light treatment for 50 days [18,19,20]. The different kinds of hormone-related genes and their sequences in *A. pernyi* are still unknown. In a study that cloned *PTTH* in *A. pernyi*, Western blotting and Northern blotting showed that the PTTH protein and mRNA are co-localized in L-NSC III cells in the brain [21]. Photoperiod is a key factor that influences diapause in *A. pernyi*. It is not clear what kind of hormones are released at a certain time under long or short photoperiod treatment. In this study, we focused on the changes in three hormones at the transcriptional level and the variation in the titer of one non-protein hormone during diapause termination and pupal development in *A. pernyi*. Information on hormonal changes in coordination with different pupal development stages will provide a better understanding of pupal diapause and guide artificial induction of diapause termination.

## 2. Materials and Methods

### 2.1. Insects

The bivoltine oak silkworm strain Shenhuang No. 2 was provided by the Research Institute for Tussah in Shenyang Agricultural University (Shenyang, China). The larvae were exclusively fed fresh oak leaves (*Quercus wutaishanica*) until they formed cocoons in autumn. Individuals in the early pupal stage were selected for the experiment on the same day they emerged from prepupae. When the pupae entered diapause, they were subjected to two photoperiod treatments. A long photoperiod treatment was established using a light incubator set under the following conditions: a 17 h:7 h light/dark period, 22 °C temperature, 60 ± 10% relative humidity, 4000 Lx illumination. A short photoperiod with a 12 h:12 h light/dark period was used as the control. Tissues (especially the brain and tracheae) were dissected from pupae every 7 days after different treatments. All samples were stored at −80 °C for later use.

### 2.2. Respiratory Intensity of A. pernyi Pupae

Insect respiration intensity was measured using 20 male and 20 female pupae to determine when the pupae enter and terminate diapause. Under natural light at room temperature, the respiratory metabolism of pupae was checked every 3 days in the first 30 days. When the respiration intensity decreased to below a certain level, respiration intensity was tested every day and lasted for many days (content of CO_2_ reduced by half). Then, the pupae were placed in the two incubators described above. The pupae were then tested every 3 days until eclosion. The concentration of carbon dioxide released per hour from each *A. pernyi* pupa (mg/m^3^) was calculated using the following formula: mg/m^3^ = (M/22.4) × [273/(273 + T)] × [P/101325] × ppm, where M is the molecular weight of CO_2_, T is the temperature (22.6 °C on average), P is the pressure (the same atmospheric pressure with no change), and PPM is the volumetric concentration change in CO_2_.

### 2.3. Total RNA Extraction, cDNA Synthesis, and Gene Cloning

For gene cloning, total RNA was extracted from whole pupae using RNAiso Plus reagent (Takara, Dalian, China), and the concentration of the total RNA was determined by spectrophotometry (NanoDrop 2000; Thermo Scientific, Waltham, MA, USA). Single-stranded cDNA was synthesized using the PrimeScript™ One Step RT-PCR Kit (Takara, Dalian, China). Primers were designed based on a transcriptome library constructed by our laboratory [22] and a genome sequence (GCA_015888305.1) using Primer Premier 6 software (Table 1). PCR amplification was programmed for an initial 3 min delay at 95 °C, 34 cycles of denaturation at 95 °C for 30 s, and annealing at 55 °C for 30 s and 72 °C for 30 s, followed by a final extension at 72 °C for 5 min. The PCR products were separated on a 1.2% agarose gel by electrophoresis and purified using the GenElute™ Gel Extraction Kit (Sigma-Aldrich, St. Louis, MO, USA). The products were cloned into the pGEM^®^-T easy cloning vector (Promega, Madison, WI, USA) and transformed into *Escherichia coli* DH5α competent cells (Takara, Dalian, China). Positive clones were selected by PCR and confirmed by sequencing (Sangon Biotech Co., Ltd., Shanghai, China).

### 2.4. Sequence Alignment and Bioinformatics

The amino acid sequences of PTTH, ETH, and EH in *A. pernyi* were determined from the deduced ORF sequences, and similar sequences from other lepidopterans were retrieved from NCBI GenBank (https://www.ncbi.nlm.nih.gov/; accessed on 1 June 2021). The theoretical isoelectric point (pI) and molecular weight (Mw) of the mature proteins were computed using the pI/Mw tool at (https//web.expasy.org/compute_pi/; accessed on 1 June 2021) Conserved domains were predicted using NCBI website tools (http://www.ncbi.nlm.nih.gov/Structure/cdd/wrpsb.cgi/; accessed on 1 June 2021). SignalP 5.0 webservers were used for locating the signal peptide cleavage sites [23]. The deduced amino acid sequences were aligned with those of other insect species using Clustal X version 2.0 [24]. A neighbor-joining (NJ) phylogenetic tree was generated with MEGA5 [25].

### 2.5. Real-Time Quantitative PCR Analysis

To evaluate hormone expression during diapause termination and eclosion, the pupae were dissected for gene expression analysis by real-time quantitative PCR (qPCR) [26]. Primers used for qPCR profiling are provided in Table 1. Gene expression was assessed using SYBR^®^Premix Ex Taq™ II (TaKaRa, Shiga, Japan) on a CFX Connect™ Real-time PCR Detection System (Bio-Rad, Hercules, CA, USA). The qPCR mixture contained 10 μL of SYBR^®^Premix Ex Taq™ II, 0.4 μL of forward primer (200 nM), 0.4 μL of reverse primer (200 nM), 2 μL of diluted cDNA, and 7.2 μL of sterile water. The thermal cycling conditions were as follows: initial denaturation at 95 °C for 30 s, followed by 40 cycles at 95 °C for 5 s and 60 °C for 30 s. For the analysis of PTTH and EH expression, RNA from the brain was extracted for qPCR. The level of ETH was measured only in the tracheae. The qPCR was performed with three independent biological replicates. The relative quantification results were normalized to the constitutively expressed β-actin gene as the reference and analyzed using the comparative C_T_ (ΔΔC_T_) method [27].

### 2.6. 20-Hydroxyecdysone Titers in Diapausing Pupae under the Long Photoperiod

The 20-hydroxyecdysone (20E) titers in the hemolymph were determined every 7 days using three pupae (0.2 mL for each pupa). Then, 20E was extracted from the pupae with 75% methanol at 4 °C overnight, followed by centrifugation at 8000× *g* for 10 min. The supernatant was dried in nitrogen and resuspended in 0.5 mL of methanol. Ten microliters of the sample was then injected into an HPLC system (Rigol L3000; Rigol Co., Ltd., Beijing, China). A Kromasil column (250 mm × 4.6 mm) packed with Lichrosorb reversed phase C−18 (5 μm) was used. The sample was eluted with 50% methanol at a flow rate of 500 μL/min at 30 °C. The fractions were collected every 30 s for 40 min, and the detection wavelength was 254 nm.

### 2.7. Data Analysis

Statistical analyses were performed using SPSS 21.0 software (IBM, Chicago, IL, USA). A Student’s *t*-test was used to compare relative expression levels. Data are presented as mean ± standard deviation of three biological replicates. Significant differences were determined using Duncan’s multiple range tests. The results were plotted using GraphPad Prism 6 (GraphPad, SanDiego, CA, USA).

## 3. Results

### 3.1. Morphological Changes during Pupal Development

From pupal diapause termination to eclosion, development can be evaluated by observing a pigment-free region of the cuticle above the pupal brain, often called a window. In diapausing pupae, the window is transparent. When the pupae approach eclosion, the window turns milky white and then red (Figure 1). The pupal body also undergoes changes during development. Diapausing pupae have more fat bodies in the tissues; the fat content decreases, and other organs emerge as the pupae approach eclosion. Additionally, eggs develop in the ovaries of females (Figure 2A). The brain, prothoracic glands (PGs), and tracheae in pupae are the main organs that release protein or non-protein hormones (Figure 2B–D).

### 3.2. Respiratory Intensity of A. pernyi Pupae

The respiratory intensity of pupae was measured to determine whether the pupae entered diapause. The average respiratory intensity was 4.51 mg/m^3^ during the first 30 days (tested every 3 days) and 2.15 mg/m^3^ from 31 to 36 days (tested every day). In the first 30 days after placing the pupae in the incubators, the two groups exhibited almost the same level of low respiratory intensity with no significant difference. On day 33, the respiratory intensity of the long photoperiod group was higher than that of the short photoperiod group (shows significant difference, *p* < 0.001). The respiratory intensity of the long photoperiod group continued to increase until eclosion, reaching 25 mg/m^3^, almost 11.62 times higher than that in the diapause period (Figure 3).

### 3.3. Sequence of ApPTTH, ApEH, and ApETH

ApPTTH (GenBank accession number: KT225461) contains a 666-bp ORF. It encodes a putative protein of 221 amino acids, 25.96 kDa Mw, and 8.38 pI. No signal peptide was found in ApPTTH. The gene has four exons and three introns in the protein-coding area of the 2491-bp DNA sequence (Appendix A). The ORF of ApETH is 345 bp (GenBank accession number: MW677186) and encodes a putative protein of 114 amino acids, 13.31 kDa Mw, and 8.51 pI. The gene has three exons in the 1166-bp DNA sequence for protein synthesis (Appendix A). The ApEH ORF (GenBank accession number: MW677185) is 267 bp and encodes a putative protein of 88 amino acids, 95.8 kDa Mw, and 4.93 pI. There is only one intron in the 7484-bp gene between two exons (Appendix A). SignalP 5.0 analysis revealed that ApETH and ApEH have an N-terminal signal peptide of 23 and 26 residues, respectively.

### 3.4. Sequence Homology Alignment and Phylogenetic Relationships of ApPTTH, ApETH, and ApEH

The homology analysis revealed that the predicted ApPTTH amino acid sequence (ALL42053) had a sequence identity of 98.19% with the PTTH sequence of *Antheraea yamamai* (AAR23822.1), and a sequence identity of 51.14% with that of *Bombyx mori* (NP_001037349.1). According to multiple alignments (Figure 4A), four lepidopteran PTTH sequences were conserved, particularly in several regions of the functional domains. The homology analysis revealed that ApETH had a sequence identity of 61.54% with SeETH (Spodoptera exigua) and 54.39% with BmETH (Figure 4B). The ApEH amino acid sequences had a sequence identity of 85.8% with those of four species (Figure 4C).

The NJ phylogenetic tree revealed that the selected amino acid sequences were divided into three groups (Figure 5). PTTH, ETH, and EH in several selected moth species were closely related to each other and formed an independent cluster, but they were distantly located from three out-group species, namely, *Papilio xuthus* (KPI97944.1, PTTH), *Tribolium castaneum* (EFA07492.2, ETH), and *P. xuthus* (KPJ05178.1, EH).

### 3.5. Expression Profiles of ApPTTH, ApETH, and ApEH from Diapause Termination to Eclosion

The relative expression profiles of ApPTTH, ApETH, and ApEH during diapause termination and development under the 17 h:7 h light/dark period treatment were detected using qPCR. Under the long photoperiod treatment, ApPTTH expression showed a small increase at 14 days, and then decreased again. At 35 days, it showed a large increase, and peaked at 42 days, showing 9.47-fold higher expression than that at 1 day (Figure 6A). The expression of ApETH increased at 42 days and peaked at 49 days, showing 6.77-fold higher expression than that at 1 day (Figure 6B). ApEH expression started to increase at 21 days and peaked at 35 days, showing 7.49-fold higher expression than that at 1 day (Figure 6C). However, after 35 days, it quickly decreased. The control group keeps diapause and eclosion rate below 2%, and the long photoperiod treatment group eclosion rate more than 97%.

### 3.6. 20E Titers

The 20E titers of all tested samples were higher than 0.30 μg/mL. At the start of diapause, the 20E titer was 0.33 μg/mL. During the long photoperiod treatment, the titer gradually increased to 0.49 μg/mL at 14 days. It then slightly decreased before increasing to 0.52 μg/mL at 35 days. Thus, there were two peaks, one at 14 days and the other at 35 days (Figure 7). All samples after long photoperiod treatment exhibit higher content than the beginning (0 days).

## 4. Discussion

Multiple methods can be used to determine whether pupae are in diapause or not. Observing the window above the brain is an easy method to determine pupal diapause or development [28]. However, with this method, we can only determine that the pupae have entered the adult stage. Therefore, it is not suitable for pre-diapause or a short period after diapause termination, and the exact time when the pupae enter or terminate diapause is not clear. In this study, we measured respiratory intensity to determine whether the pupae had entered a low metabolic rate. We found that the pupae enter diapause at approximately 1 month after pupal emergence. In a previous study, post-diapause adult (31.6 ± 4.4 days, *N* = 31) of a bivoltine strain of *A. pernyi* emerged at 25 °C under 16 h L:8 h D photoperiod [29]. Here, low respiratory intensity detected both in two groups within 1 month after photoperiod treatment, but the exact time of diapause termination could not be detected from these results.

The brain and PGs are the two main organs that control insect larval and pupal changes [30]. PTTH is a regulator of PG ecdysteroidogenesis (ecdysteroid biosynthesis). Ecdysteroids are molting hormones (MH), including ecdysone and 20E, which are mainly produced by PGs and are essential for molting and metamorphosis. Moreover, 20E is produced from the ecdysone by P450 monooxygenase, which is encoded by Halloween genes [31,32]. Several steps in 20E synthesis remain unknown, and they are referred to as “black box” steps. As a non-protein hormone, 20E triggers insect diapause termination. In this study, the relative expression of *ApPTTH* slightly increased at 14 days under the long photoperiod treatment, and then increased to a greater extent at 35 days. A similar pattern was observed for the 20E titer in the hemolymph. The result suggests that PTTH is an important regulator of 20E release in PGs of *A. pernyi*.

The sequences of two more new genes, *ApETH* and *ApEH*, were obtained in this study. EH is a neuropeptide released into the CNS and hemolymph [33]. It plays a direct or indirect role in the abdominal neural network. Initially, only EH was thought to be responsible for triggering ecdysis [34]. Later, a second ecdysis-triggering peptide, ETH, was found to have a complex relationship with EH [11]. ETH acts directly on the central nervous system (CNS) to coordinate muscle contractions during the ecdysis phase [13]. Here, the homology analysis revealed that EH is more conserved in insects than ETH. In *M. sexta* and *B. mori*, EH has three intramolecular disulfide bonds and a conserved secondary structure [35,36]. ApEH account for 85.8% identity with other three EH sequences, and it is more conserved than ETH. This indicates that EH is conserved in evolution and may be have basic functions among different insect species. The isolation of EH provided the first indication that insect ecdysis is under hormonal control [37]. A lack of EH function is completely lethal in *Drosophila* [14]. EH also promotes ETH release for the final shedding of the old cuticle [38]. In other insects, EH has been found to have other biological functions, for example, regulating the release of the hormone bursicon in *M. sexta* [39]. In our study, *ApEH* expression was high from 21 to 35 days, but it was low in the ecdysis phase. *ApETH* began to increase later under the long photoperiod, just before adult eclosion. Thus, ETH may not have a direct influence on EH release. Previous studies have demonstrated that 20E can also regulate ETH synthesis and release [40]. *EH* or *ETH* may be interrelated and may influence each other in a complex way. These findings indicate that *ApEH* synthesis may activate *ApETH* during the diapause termination process, especially in the cuticle shedding stage.

Diapause is a complex process in different insect species. It is often associated with the changes in hormone and carbohydrate metabolism. They are important influence factors and can induce insects to enter or terminate diapause. Previously, we evaluated the changes in trehalose level and the related genes during diapause in *A. pernyi* [18,19]. In this study, we focused on hormone expression patterns. The following questions remain unanswered: Which cells release the hormones? What are the hormone receptors? What is the target tissue? For example, after PTTH secretion into the hemolymph, PTTH binds to a receptor located at the surface of PGs [41]. A receptor tyrosine kinase (RTK), named Torso, is the PTTH receptor [42]. In *A. pernyi*, a *Torso*-like gene was found to have 74% identity with *B. mori Torso* (NP_012546780) [43]. The ETH receptor (ETHR), a typical G protein-coupled receptor, has two functionally distinct isoforms. Being result of alternative splicing, there are *ApETHR-A* and *ApETHR-B* in *A. pernyi* (data not published). Further research is required to determine which one is required for diapause termination or ecdysis.

## 5. Conclusions

Respiratory intensity was tested to determine the exact time when the pupae entered diapause. Under long photoperiods, pupae will end diapause in almost 1 month, and their respiratory intensity rapidly increases compared with that in the short photoperiod group. One hormone titer and three hormone-related gene expressions were explored at the molecular level. These changes may be key factors that influence the development process. These findings will help us in understanding the molecular mechanism in pupal diapausal insects. Identifying the reason for entering diapause could enhance breeding of multi-generation tussah pupae in areas where they are naturally univoltine.

## Figures and Tables

**Figure 1 insects-12-00699-f001:**
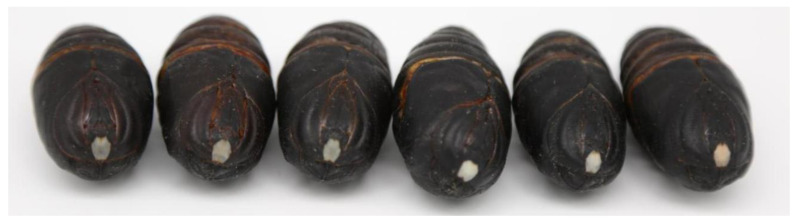
Changes in the window above the pupal brain during diapause and eclosion. From diapause to eclosion, the window changes from transparent to milky white, and finally to red.

**Figure 2 insects-12-00699-f002:**
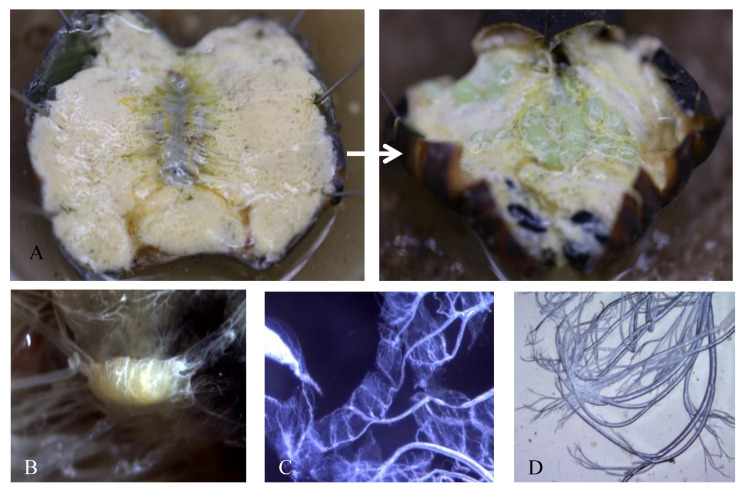
Morphological changes in the pupae: (**A**) Changes in the bodies of pupae, with respect to fat body, gut, and eggs (female). Before development, the whole body is full of fat, but after development, eggs emerge, and fat bodies reduce considerably. (**B**) Pupal brain. (**C**) Pupal prothoracic gland. (**D**) Pupal tracheae.

**Figure 3 insects-12-00699-f003:**
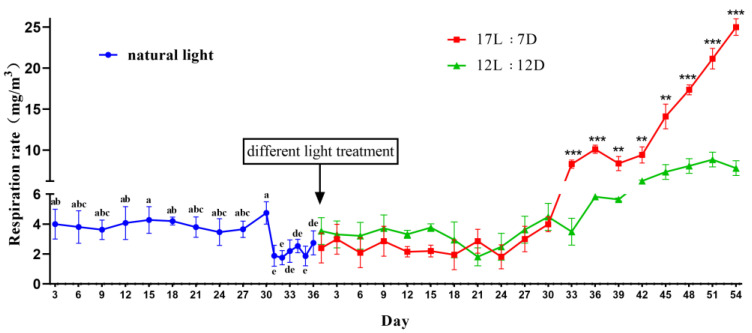
Respiratory metabolism from pupation to diapause, and eclosion under a long photoperiod (17 h L:7 h D) compared with that under a short photoperiod (12 h L:12 h D; control). Data are means ± SD. The significant differences under natural light period were determined using Duncan’s multiple range tests. Different letters above bars show significant difference (*p* < 0.05). Under two photoperiod treatment, significant differences was determined using Student’s *t*-test (** *p* < 0.01; *** *p* < 0.001).

**Figure 4 insects-12-00699-f004:**
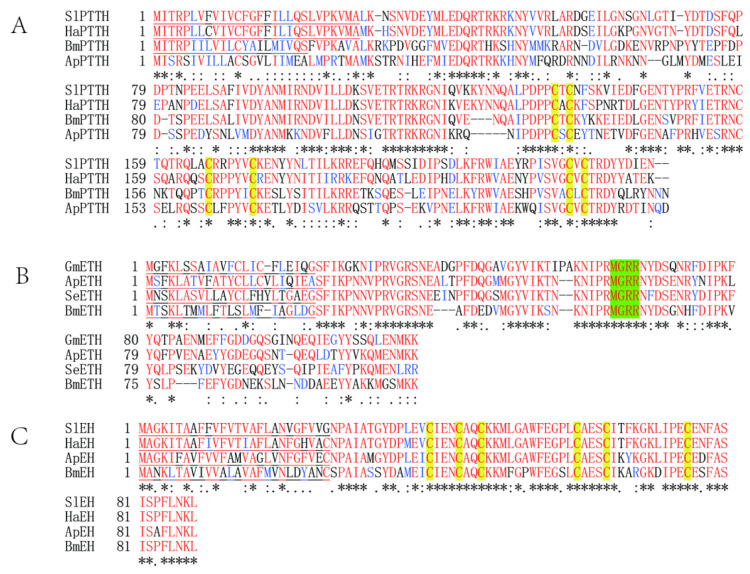
Sequence alignment of the deduced amino acid sequences of PTTH (**A**), ETH (**B**), and EH (**C**). Species abbreviations and accession numbers of the sequences used in the alignment are as follows: SlPTTH—*Spodoptera litura* (XP_022816485.1), HaPTTH—*Helicoverpa armigera* (AAP41131.1), BmPTTH—*Bombyx mori* (NP_001037349.1), ApPTTH—*Antheraea pernyi* (ALL42053.1); GmETH—*Grapholita molesta* (QMS43279.1), SeETH—*Spodoptera exigua* (AXY04252.1), BmETH—*Bombyx mori* (NP_001165743), ApETH—*Antheraea pernyi* (MW677186), SlEH—*Spodoptera litura* (XP_022825948.1), HaEH—*Helicoverpa armigera* (XP_021186221.1), BmEH—*Bombyx mori* (XP_012553269.), and ApEH—*Antheraea pernyi* (MW677185). Identical and similar amino acid residues are marked with asterisk, colon, and dot. Predicted signal peptides are underlined. Conserved cysteine (**C**) residues involved in disulfide bridge are shaded in yellow in PTTH and EH. The amino acids required for amidase activity in ETH are shaded in green.

**Figure 5 insects-12-00699-f005:**
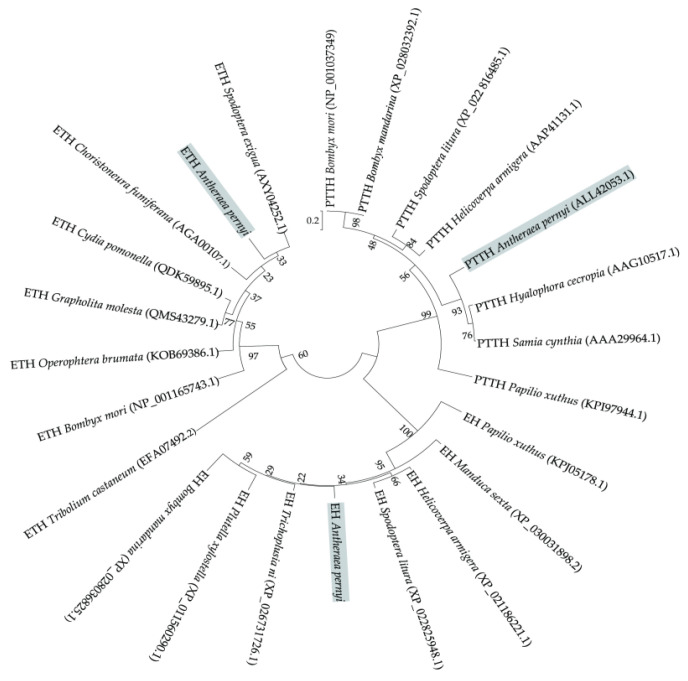
Phylogenetic tree constructed using PTTH, ETH, and EH sequences using MEGA7.0. Bootstrap percentage values are shown on the branches. GenBank accession numbers, along with organism names, are shown for all the sequences. *Papilio xuthus* (KPI97944.1), *Tribolium castaneum* (EFA07492.2), and *Papilio xuthus* (KPJ05178.1) were used for comparison as out-groups.

**Figure 6 insects-12-00699-f006:**
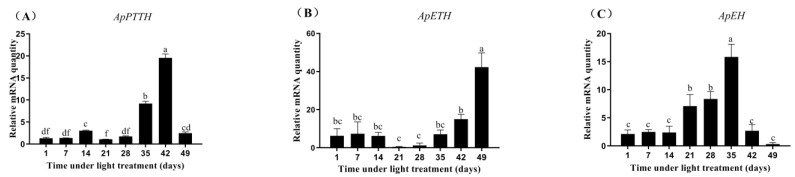
Relative expression levels of *ApPTTH* in the brain (**A**), *ApETH* in the tracheae (**B**), and *ApEH* in the brain (**C**) of *Antheraea pernyi* pupae during diapause termination and development detected by qPCR. Pupae were exposed to a long photoperiod (17 h L:7 h D) and sampled every 7 days. This experiment was conducted in three replicates; the bar represents mean ± SE (n = 3) and different letters above the bars show significant differences determined using Duncan’s multiple range test (*p* < 0.05).

**Figure 7 insects-12-00699-f007:**
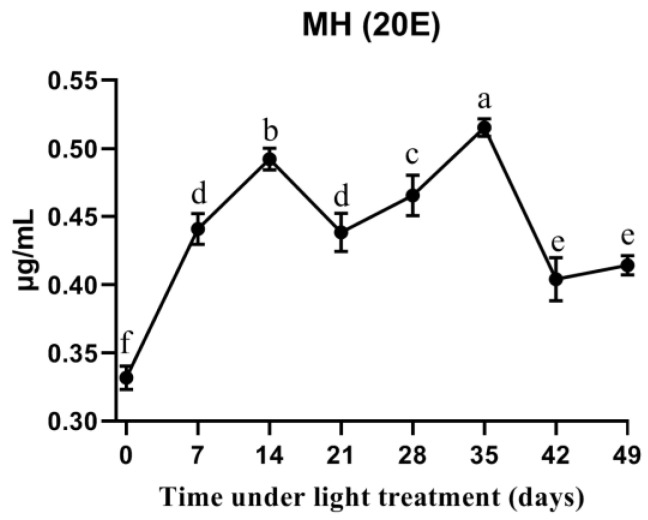
Ecdysteroid (20E) titer changes in pupal hemolymph from diapause to eclosion under the long photoperiod (17 h L:7 h D).

**Table 1 insects-12-00699-t001:** Oligonucleotide primers for this study.

Gene Name	Primer Name	Primer Sequence (5′–3′)	For Purpose
ApPTTH	ApPTTH-F	TCCTGGAGGTTCAAACAACACAT	for RT-PCR
	ApPTTH-R	CTACCAAAAACAAAGCATTAGTATCAT	for RT-PCR
ApETH	ApETH-F	ACGTTTCGAAAGTTTTAATCATGAGT	for RT-PCR
	ApETH-R	TGTACCGGTTTTCCGAATCATAAT	for RT-PCR
ApEH	ApEH-F	ATGGCCGGCAAGATCTTCG	for qPCR
	ApEH-R	TTAGAGTTTGTTGAGGAAGGCTGAG	for qPCR
q-ApPTTH	q-ApPTTH-F	ACAATCATCGTGTCTCTTTCCGT	for qPCR
	q-ApPTTH-R	TTTTCAGCAATCCACCTAAACTTC	for qPCR
q-ApETH	q-ApETH-F	CAACATACTGCCTACTCTGCGTCC	for qPCR
	q-ApETH-R	CAAAGCCTCGTTGCTCCGTC	for qPCR
q-ApEH	q-ApEH-F	ATCTTCGCTGTTTTCGTTGTTTT	for qPCR
	q-ApEH-R	TTCCAGAGGATCGTAGCCCAT	for qPCR
q-β-actin	q-β-actin -F	ACCAACTGGGACGACATGGAGAAA	for qPCR
	q-β-actin-R	TCTCTCTGTTGGCCTTTGGGTTGA	for qPCR

## Data Availability

The data presented in this study can be found in the Appendix A

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
