# Peer review of "Expression Patterns of Three Important Hormone Genes and Respiratory Metabolism in Antheraea pernyi during Pupal Diapause under a Long Photoperiod"

_insects, 2021, doi:10.3390/insects12080699_

Round 1
Reviewer 1 Report
I found the manuscript potentially interesting. The species (Antheraea pernyi) can be univoltine and bivoltine in China, according to the climatic patterns; consequently, the climatic conditions interferes with the diapausing process…
However, I found a big discrepancy between the quality of the basic biological data and bioassays and the bio-molecular part: the firsts have several criticisms while the seconds were following a more solid and appropriate approach. Some of the experiments carried out to get basic biological data should be improved: they are not very well designed and very often there is not any statistical analysis.
I also found the Discussion and -in particular- the Conclusion chapters with a very limited final analysis of the achieved results of the work done. The feeling is taht the authors did not know what are the results of the work done, what are the possible criticisms and what should be the next steps. Finally, the English needs an important improvement.
My suggest is to ask the authors to make an important revision of the manuscript, enhancing the sub-chapters 1, 2 and 6 (both for Mat & Met and for Results); and improve eventually a little the Introduction (It is not clear the reason why they are studying this species) and definitely Discussion and Conclusions.
Author Response
I found the manuscript potentially interesting. The species (Antheraea pernyi) can be univoltine and bivoltine in China, according to the climatic patterns; consequently, the climatic conditions interferes with the diapausing process…
However, I found a big discrepancy between the quality of the basic biological data and bioassays and the bio-molecular part: the firsts have several criticisms while the seconds were following a more solid and appropriate approach. Some of the experiments carried out to get basic biological data should be improved: they are not very well designed and very often there is not any statistical analysis.
Response: Thank you for your suggestion. Some of data for biological study are not good as bio-molecular research. For 20E titer study, we sent three samples for testing again. At the very beginning, some samples we test triple, small errors in their expression. We only test one sample in the later experiment in order to save money. In this time, other test have been done. For respiratory intensity, because we only have one machine and every time it will consume a long time. Every day we chosen a regular time to test samples. If test for three times, we must do that in different times (light and temperature may influence respiratory intensity) in different stage. That would influence each group in a day. At right now, we not have diapause pupae to repeat this experiment.
I also found the Discussion and -in particular- the Conclusion chapters with a very limited final analysis of the achieved results of the work done. The feeling is that the authors did not know what are the results of the work done, what are the possible criticisms and what should be the next steps. Finally, the English needs an important improvement.
Response: We have changed the discussion and conclusion chapter. The English have been edit and polished by editage company.
My suggest is to ask the authors to make an important revision of the manuscript, enhancing the sub-chapters 1, 2 and 6 (both for Mat & Met and for Results); and improve eventually a little the Introduction (It is not clear the reason why they are studying this species) and definitely Discussion and Conclusions.
Response: We have revised abstract, discussion and conclusions totally. And make some revisions in Introduction, Mat and results.
Reviewer 2 Report
This manuscript reports on some of the endocrine and metabolic features that accompany diapause and the end of diapause in the moth, Antheraea pernyi. It also reports the sequence of the hormones, PTTH. EH, and ETH for this species. Diapause was broken using a long day photoperiod, and was accompanied by an increase in respiration rate, in 20E titers, and in the expression of PTTH, EH, and ETH. The sequences of PTTH, EH, and ETH are similar to those of other lepidopteran species.
The work reported is well done, although the information it contributes is of limited significance, since it parallels what is known from other insect species, including other lepidopteran species.
I only have a few minor comments:
1- The authors present no experimental evidence that supports their statements that 20E levels are causally coordinated with PTTH expression, nor that EH can promote ETH release and that ETH triggers ecdysis. These statements are likely correct based on what we know from other insects, but they do not derive from the current work and should be eliminated from the relevant sections (Abstract and Discussion), or at the very least be presented differently.
2- In Figure 2, the “A” label of the top left panel as well as the red arrow connecting the top panels are not visible.
3- The sequences in Figure 4 are obscured by the shading used (especially the black boxes). Please use a different way to mark identities and similarities (e.g., asterisks below sequence to indicate identity, etc).
4- Figure 5: the font is too small to be readable. Please use larger letters.
5- Figure 6: letters used to indicate statistical significance are too small. Please use larger letters.
6- Line 287. Why is the comparison between the number of introns present in the PTTH, ETH and EH genes of interest? This seems like a superfluous piece of information
7- Regarding the sequences of ApPTTH, ApETH, and ApEH, please state whether they include key features of each hormone in (e.g., the key cysteines in EH and PTTH; the (putative) cleavage sites within the ETH and amidation signal at the end, etc).
Author Response
This manuscript reports on some of the endocrine and metabolic features that accompany diapause and the end of diapause in the moth, Antheraea pernyi. It also reports the sequence of the hormones, PTTH. EH, and ETH for this species. Diapause was broken using a long day photoperiod, and was accompanied by an increase in respiration rate, in 20E titers, and in the expression of PTTH, EH, and ETH. The sequences of PTTH, EH, and ETH are similar to those of other lepidopteran species.
The work reported is well done, although the information it contributes is of limited significance, since it parallels what is known from other insect species, including other lepidopteran species.
I only have a few minor comments:
- The authors present no experimental evidence that supports their statements that 20E levels are causally coordinated with PTTH expression, nor that EH can promote ETH release and that ETH triggers ecdysis. These statements are likely correct based on what we know from other insects, but they do not derive from the current work and should be eliminated from the relevant sections (Abstract and Discussion), or at the very least be presented differently.
Response: Thank you for your suggestions. We think this statement is not suitable here. So these discussions all delete in the abstract and discussion.
- In Figure 2, the “A” label of the top left panel as well as the red arrow connecting the top panels are not visible.
Response: Have been revised.
- The sequences in Figure 4 are obscured by the shading used (especially the black boxes). Please use a different way to mark identities and similarities (e.g., asterisks below sequence to indicate identity, etc).
Response: Have been revised. We have changed the shading style. Using colored number with asteriskes mark identities.
4- Figure 5: the font is too small to be readable. Please use larger letters.
Response: Have been revised.
5- Figure 6: letters used to indicate statistical significance are too small. Please use larger letters.
Response: Have been revised.
- Line 287. Why is the comparison between the number of introns present in the PTTH, ETH and EH genes of interest? This seems like a superfluous piece of information
Response: Yes, I agree with you. These seems no use for this study. We delete it.
7- Regarding the sequences of ApPTTH, ApETH, and ApEH, please state whether they include key features of each hormone in (e.g., the key cysteines in EH and PTTH; the (putative) cleavage sites within the ETH and amidation signal at the end, etc).
Response: These analysis of putative cleavage sites and amidation have perform this time. Because we think these analysis may be not important for diapause study in this research. If definitely needed, we will add that later.